# Utility of Verification Testing to Confirm Attainment of Maximal Oxygen Uptake in Unhealthy Participants: A Perspective Review

**DOI:** 10.3390/sports9080108

**Published:** 2021-07-30

**Authors:** Todd A. Astorino, Danielle Emma

**Affiliations:** Department of Kinesiology, California State University, San Marcos, CA 92096, USA; emma001@cougars.csusm.edu

**Keywords:** verification testing, maximal oxygen uptake, unhealthy adults, graded exercise testing, VO_2_max criteria

## Abstract

Maximal oxygen uptake (VO_2_max) is strongly associated with endurance performance as well as health risk. Despite the fact that VO_2_max has been measured in exercise physiology for over a century, robust procedures to ensure that VO_2_max is attained at the end of graded exercise testing (GXT) do not exist. This shortcoming led to development of an additional bout referred to as a verification test (VER) completed after incremental exercise or on the following day. Workloads used during VER can be either submaximal or supramaximal depending on the population tested. Identifying a true VO_2_max value in unhealthy individuals at risk for or having chronic disease seems to be more paramount than in healthy and active persons, who face much lower risk of premature morbidity and mortality. This review summarized existing findings from 19 studies including 783 individuals regarding efficacy of VER in unhealthy individuals to determine its efficacy and feasibility in eliciting a ‘true’ VO_2_max in this sample. Results demonstrated that VER is a safe and suitable approach to confirm attainment of VO_2_max in unhealthy adults and children, as in most studies VER-derived VO_2_max is similar of that obtained in GXT. However, many individuals reveal higher VO_2_max in response to VER and protocols used across studies vary, which merits additional work identifying if an optimal VER protocol exists to elicit ‘true’ VO_2_max in this particular population.

## 1. Introduction

Maximal oxygen uptake (VO_2_max) as determined by the Fick Equation represents the maximal ability of the cardiovascular system to transport oxygen and the capacity of the periphery to extract oxygen to support aerobic metabolism. It is apparent that VO_2_max is related to endurance performance and, more importantly, premature mortality [1]. Because of this link between VO_2_max and health status, the American College of Sports Medicine [2] recommends 150 min/week of moderate intensity continuous exercise or 75 min/week of vigorous exercise to enhance fitness and improve overall health status, although attainment of this guideline in U. S. adults is relatively low [3].

Despite the fact that VO_2_max has been measured in laboratory and clinical settings for a century, there is no standardized exercise testing protocol to assess it as the specific work rate increment, stage duration, and gas exchange sampling interval vary across studies. In addition, there is no robust approach to ensure that VO_2_max is attained at the end of incremental exercise which is problematic when this value is used to prescribe exercise, assess training responsiveness, or describe health status. In turn, relying on an imprecise estimate of VO_2_max can have negative effects upon the accuracy of these applications which can change the course of decision making made by practitioners or scientists regarding client health. Various primary (oxygen plateau) and secondary criteria (maximal values of heart rate, respiratory exchange ratio, rating of perceived exertion, and blood lactate concentration) are widely used in this capacity, yet each has its limitations (for additional information on this, please consult Schaun et al. [4]) that may make them ineffective in ensuring that VO_2_max is actually attained by each participant.

Implementation of a second exercise test completed after the incremental test was first identified by Thoden et al. [5] in active adults who required an ‘exhaustive test’ to be performed after the incremental protocol. Later work [6,7] showed that completion of this subsequent higher intensity bout (called the verification test (VER)), performed a few minutes or up to 1 week after the incremental exercise bout, led to similar mean estimates of VO_2_max, thus confirming a plateau in oxygen uptake and, in turn, attainment of VO_2_max. For example, in 16 distance runners, data [8] showed that 26 of 32 VO_2_max tests performed on a treadmill reveal similar (≤2% different) estimates of VO_2_max between ramp and subsequent verification testing. In seven healthy men, Rossiter et al. [9] demonstrated that VER at 95 or 105%of peak power output (PPO) performed 5 min after ramp exercise elicits similar values of VO_2_max, leading these authors to recommend either protocol as a suitable way to confirm VO_2_max attainment. Overall, these data show that VER is a robust procedure to confirm attainment of VO_2_max in healthy active adults.

Despite these data, a valid concern of VER is that its supramaximal effort would be inappropriate for those who are inactive or at risk for chronic disease who lack the exercise capacity due to aging, presence of comorbidities, or desire to sustain such demanding efforts long enough to allow VO_2_ to attain a maximal value. However, results from inactive adults [10], older adults [11], and those with obesity [12,13,14] demonstrate that it is well-tolerated and feasible in these populations and leads to similar estimates of VO_2_max as the ramp test. In addition, data show its efficacy to confirm attainment of VO_2_max in adults with metabolic syndrome [15] as well as heart failure [16]. Recent data also show that implementing VER reveals more precise determinants of increases in VO_2_max in response to high intensity interval training in adults with metabolic syndrome compared to graded exercise testing [17]. So, similar to healthy adults, use of VER seems warranted to confirm attainment of ‘true’ VO_2_max in persons with chronic disease.

A recent systematic review [18] summarized data concerning efficacy of VER in healthy participants and concluded that this is a robust approach to confirm the value acquired from incremental exercise. However, having a more accurate estimate of ‘true’ VO_2_max in this active population may not be that important as their cardiorespiratory fitness is superior, leading to enhanced health status versus less fit populations. In response to exercise training, an increase in VO_2_max as low as 1.5 mL/kg/min has been identified as being clinically significant in persons with chronic disease [19]. Thus, in persons having low VO_2_max and, in turn, diminished health status, any small inaccuracies in VO_2_max assessment may elicit different responses to training and/or inaccurate diagnoses that may modify choice of various treatment options implemented to improve individual health status. In addition, VO_2_max is frequently measured as a primary outcome in exercise training studies due to its strong relationship with health status [1]. Moreno-Cabanas et al. [17] concluded that ramp testing misrepresents the training-induced change in VO_2_max in a majority of individuals with metabolic syndrome, and they emphasized the necessity of VER to better represent the VO_2_max response to training. However, to our knowledge, no review article has summarized efficacy of VER to confirm VO_2_max incidence in unhealthy participants. Some studies show that VER leads to similar estimates of VO_2_max versus graded exercise testing, whereas others show significantly higher VO_2_max when VER is performed. These equivocal findings may cloud judgment as to whether this additional test should be performed to elicit a ‘true’ VO_2_max and merit development of a review article to provide a thorough overview of feasibility of VER in clinical populations.

This review summarized findings regarding efficacy of VER to confirm attainment of VO_2_max in unhealthy and/or inactive participants which, to our knowledge, has not been done. The main questions answered by this review include: (1) is verification testing able to confirm attainment of VO_2_max in this sample, (2) is it safe and well-tolerated, and (3) is there an optimal intensity or structure of VER to employ to confirm attainment of VO_2_max in this particular sample? Results from Murias et al. [20] obtained in young and older men concluded that VER is unnecessary to confirm VO_2_max attainment as mean VO_2_max values from this test and the preceding ramp test were not significantly different. Recent work from this laboratory [21] also revealed that VER using supramaximal workloads significantly underestimated VO_2_max, so these authors did not recommend these intensities for VER testing. Nevertheless, these results were acquired in active adults that do not apply to individuals with lower cardiorespiratory fitness. Moreover, no individual results were presented which is important since attaining a ‘true’ VO_2_max is an individual phenomenon. Recent work in adults with cancer [22], hypertension [23], and obesity [13] reveal that a sizable amount of individuals exhibit an underestimation in ramp-derived VO_2_max and a higher VO_2_max value when supramaximal VER is performed, which supports its efficacy in inactive individuals. However, across all studies, the participant population, testing protocol used, and criteria employed to confirm VO_2_max incidence vary, which does not allow identification of a standard VER protocol in clinical populations. Overall, detecting a ‘true’ VO_2_max is paramount, as this value can be used to prescribe personalized exercise training, assess efficacy of exercise training, and classify health risks.

## 2. Materials and Methods

### 2.1. Search Strategy

We conducted a literature search from February to April 2021 using databases including PubMed, Google Scholar, and SPORTDiscus. The key words used were ‘maximal oxygen uptake,’ ‘VO_2_max,’ ‘maximal oxygen consumption,’ AND ‘verification testing,’ and ‘supramaximal.’ Additional articles were also identified by using the references lists of selected articles. Inclusion criteria were studies written in English using incremental exercise testing leading to VO_2_max followed by verification testing to confirm attainment of VO_2_max at submaximal up to supramaximal intensities. In addition, studies using participants who have or are at risk for chronic disease were included, which encompassed the following populations: inactive adults or children; adults with obesity; older adults >50 years; and adults or children with underlying disease including cancer, diabetes, cardiovascular disease, etc. These criteria were chosen as a recent review paper extensively summarized the efficacy of verification testing in healthy adults [18]. Studies were excluded if submaximal protocols were used to assess VO_2_max, as well as those not acquiring gas exchange data.

### 2.2. Outcomes Identified

From each article, we extracted the following information: The traits of the participants including age, health status, physical activity status, and body mass index, which was calculated from height and mass if not presented. In addition, we denoted the exercise modality completed, as well as the specific traits of both the incremental and verification test as well as the recovery duration between these tests. As far as the physiological outcomes, we identified the relative VO_2_max from each protocol, as well as HRmax and test duration of the incremental and verification test.

### 2.3. Data Analysis

Results are presented as mean ± SD where appropriate.

## 3. Results

### 3.1. Summary of Studies

Table 1 presents a summary of the 19 studies included in this review, consisting of 783 adult men and women and children. The populations included in these studies were children or older adults (*n* = 2) who are inactive (*n* = 2), overweight or obese (*n* = 5), had cancer (*n* = 1), congestive heart failure (*n* = 1), metabolic syndrome (*n* = 1), hypertension (*n* = 2), cystic fibrosis (*n* = 3), spina bifida (*n* = 1), or had spinal cord injury (*n* = 2). Across participants, age ranged from preadolescent up to adults over 60 years of age. Seven studies contained participants who were inactive, and five studies had participants who were recreationally active. Eleven studies included participants with BMI values greater than 24.9 kg/m^2^, whereas seven studies included participants with BMI below this value.

### 3.2. Methods Used to Assess VO_2_max during Incremental and Verification Testing

Table 2 denotes the methods used to assess VO_2_max from graded exercise testing and the subsequent verification test. Fourteen studies utilized primary (VO_2_ plateau) and secondary criteria (RERmax, HRmax, RPE, and/or blood lactate concentration) to verify attainment of VO_2_max from GXT, although five studies did not report that any VO_2_max criteria were used. Cycling was the modality used in 14 of 19 studies, with 1 study employing arm ergometry [23] and 4 studies using treadmill exercise in overweight to obese adults [12], adults with hypertension [22], athletes with spinal cord injury [24], and children with spina bifida [25]. The most widely used protocol to assess VO_2_max during GXT was a traditional ramp test (*n* = 10 studies), although in nine studies, a step incremental test was used. Studies were characterized by various intervals between protocols, with durations as brief as four minutes to as long as a few hours between tests. Two studies required VER to be performed 24–48 h after completion of GXT.

As far as the intensity of VER, 2 studies used a submaximal protocol [16,26], 15 studies used supramaximal work rates ranging from 105–115% PPO or above maximal TM velocity, and 3 studies [12,26,27] used workloads equivalent to PPO. Eight studies included specific criteria to identify differences in VO_2_max between protocols which were developed through reliability testing or predicted changes in VO_2_ for the change in work rate.

### 3.3. Differences in VO_2_max between Ramp and Verification Testing

Table 3 denotes VO_2_max values measured in response to GXT and VER for the studies included in this review. Results from 13 of 19 studies [5,9,10,11,12,13,23,24,25,28,29,30,33] revealed no significant difference in mean VO_2_max between protocols, although in 7 of these studies [9,10,12,13,25,28,29], individual participants revealed meaningfully higher VO_2_max (≥3% higher) with VER compared to GXT. Nevertheless, in six studies [14,22,26,27,31,32] the VER-derived VO_2_max was significantly higher than GXT, with participants’ VO_2_max ranging from 19–40 mL/kg/min. In one study in cancer patients [21], VER-derived VO_2_max was significantly lower than from GXT.

### 3.4. Differences in HRmax between Ramp and Verification Testing

HRmax values from GXT and VER are demonstrated in Table 3. Similar to VO_2_max, the majority of studies exhibit no differences in maximal HR between protocols. Results from one study in obese adults [12] revealed a higher HRmax in response to VER, although another study [9] showed lower HRmax with VER versus GXT.

### 3.5. Exercise Duration of Verification Testing

Table 3 shows durations of VER reported in the studies. The shortest duration was equal to 1.5 min [13], with this VER protocol lasting up to 7 min in obese adults performing this bout at 80% PPO [26]. Twelve of nineteen studies were characterized with VER duration less than 3 min [9,12,13,14,15,21,23,26,27,28,29,32], with five studies having duration equal to or less than 2 min [12,13,15,23,28].

## 4. Discussion

Despite the widespread testing and application of VO_2_max in the fitness, clinical, and research setting, there is no universal approach to confirm its attainment from graded exercise testing. Verification testing is another widely adopted method to perform this function, yet it has been criticized for requiring an additional intense effort that may be inappropriate in those who are not active or healthy. A prior review by Poole and Jones [34] emphasized the widespread implementation of verification testing to identify a ‘true’ VO_2_max rather than ‘VO_2_peak’ in healthy active adults. In contrast, recent work [19] in active young and older men concluded that verification testing is unnecessary due to lack of differences in mean VO_2_max between the incremental and verification-derived value. The current review adds to this dogma by summarizing existing results from a large population of unhealthy adults and children completing verification testing following a GXT. Obtaining the most accurate VO_2_max value in this population is vital as it may lead to misrepresentations in their health status or responsiveness to training, which may in turn lead to inappropriate courses of treatment. Results reveal that most studies show no differences in aggregate VO_2_max between protocols. However, six studies show that VER elicits significantly higher estimates of VO_2_max, which supports its use when utmost accuracy is required in determining a ‘true’ VO_2_max on that day of testing.

Identifying differences in VO_2_max between GXT and VER requires that scientists are aware of the magnitude of error in VO_2_max estimation for both protocols. The error inherent in repeated VO_2_max testing ranges from 2–9% [7,14,28,35], with the error in acquiring gas exchange data from a metabolic cart being small (40 mL/min for the Parvo Medics system). This suggests that the remainder of the error is biological and likely related to participants’ ability and motivation to tolerate near maximal exercise. We recommend that scientists perform repeated testing to develop typical error values for their lab and use these values when comparing individual VO_2_max values between protocols rather than only comparing aggregate values. This approach, albeit time intensive, is preferred since relying on other laboratories’ criterion values is inappropriate due to differences in exercise protocol, equipment, patient population, pre-test dietary and physical activity restrictions, and time averaging intervals, which likely induce small changes in oxygen uptake.

A primary criticism of supramaximal VER testing is that this effort is too intense for inactive, unhealthy, or deconditioned adults to tolerate, resulting in a very brief duration of exercise and greater potential to not attain VO_2_max due to slow O_2_ kinetics. However, data from multiple studies [12,15,28,29] using supramaximal VER with exercise duration <2 min exhibit no differences in VO_2_max between protocols, similar to studies [9,10,29,30] in which VER duration lasted between 2–4 min. A recent study in hypertensive adults [22] used a multi-stage verification protocol eventually requiring a supramaximal workload. Results showed a significant underestimation of mean and individual VO_2_max values in response to GXT compared to VER. In nine obese adults with VO_2_max equal to 35 mL/kg/min [26], VER at 105% PPO elicited significantly lower exercise duration (167 s) compared to VER at 80% PPO (418 s), although there was no difference in VO_2_max between tests. However, VER performed at 80 (+0.16 L/min, 5% higher) and 90% PPO led to a higher VO_2_max value (+0.24 L/min, 7% higher) versus GXT, although this latter result was a trend (*p* = 0.06). Bhammar et al. [29] reported that a minimum exercise duration to attain a plateau in VO_2_ in response to VER in patients with hypertension was 80 s. These results seem to indicate that the appropriate or minimum duration required to allow attainment of ‘true’ VO_2_max using VER in unhealthy adults and children is similar to that recommended for healthy and active individuals. Thus, it is possible that submaximal intensities or multi-stage protocols may optimize VO_2_max values compared to GXT, although additional work in larger samples is needed to confirm this result.

Our review corroborates results from healthy, fit adults [17,36] showing no difference in HRmax between GXT and VER. However, a subset of data presented in this study [36] from participants with average cardiorespiratory fitness, exhibited significantly lower HRmax (−3 b/min) in response to VER compared to GXT. This is likely a result of the stepwise protocol used in this study that is characterized by a work rate less than PPO eliciting VO_2_max combined with a relatively long exercise duration (~20 min) versus the traditional 8–12 min ramp protocol. In contrast, obese adults performing VER at 100% PPO expressed significantly higher HRmax (+3 b/min) versus GXT [12], which may be attributed to their unfamiliarity with vigorous exercise during the initial incremental bout. To identify a ‘true’ VO_2_max, Midgley and Carroll [37] denoted a difference in HRmax < 4 b/min between GXT and VER. This value encompasses the magnitude of differences in HRmax described in the above studies, so it is likely that these discrepancies in HRmax between protocols are not clinically meaningful.

Considerations as to the exact characteristics of the recovery interval between GXT and VER include the intensity of the verification test, duration of GXT, cardiorespiratory fitness of participants, as well as a potential need to reduce the overall time of the session. Our review (Table 2) shows durations as brief as 2–5 min between protocols [24,25,27,30,31], 5–15 min [13,14,15,22,24,28,32,33], to as long as several days between protocols [10,27]. A recent systematic review [17] concluded that there was no effect of recovery interval on the difference in VO_2_max between protocols, which would suggest that any duration is appropriate. It is also apparent that some studies require an active recovery between protocols [13,15,23,31], whereas a passive recovery is completed in other investigations [11,14,22,25,29,32]. We recommend that scientists perform preliminary testing to identify an optimal recovery protocol for their specific population, and if this is implausible, then we recommend that they duplicate previously used procedures for that population.

Verification testing is only appropriate to identify ‘true’ VO_2_max if it is safe and well-tolerated by the participant completing exercise testing. This factor is especially critical in persons unfamiliar with vigorous exercise who may face enhanced risk of complications during vigorous exercise. In male and female survivors of cancer, Schneider et al. [21] reported no adverse events in their participants performing VER at 110% PPO. Furthermore, use of VER in adults with heart failure [15], hypertension [29], and metabolic syndrome [14] was described as “feasible” and “well-tolerated” in these populations at risk for or having heart disease. In children with cystic fibrosis [33], it was labeled as “safe.” Although further work is needed to substantiate this, empirical results suggest that VER following GXT is a safe and well-tolerated procedure that does not induce contraindications to exercise testing in persons who are inactive, have known disease, or exhibit enhanced risk of cardiometabolic disease. This guideline encompasses all VER protocols requiring efforts at submaximal, maximal, or supramaximal work rates. The only disadvantage to VER seems to be the extra time commitment required of approximately 15–20 min, including the recovery between protocols. However, this extra time is acceptable if the primary goal of testing is to acquire the most precise estimate of VO_2_max, which is critical in “at-risk” individuals when VO_2_max testing is used to identify health status or determine the effects of exercise training.

There are a few limitations to this review. First, the marked diversity in patient populations used and the specific GXT and VER protocol completed preclude us from making universal recommendations regarding an optimal verification test. Nevertheless, it seems that submaximal or supramaximal work rates can be employed with little difference in resultant VO_2_max values expected versus GXT. Second, with exception of a few studies [11,14,21,31], the sample size of individual studies is relatively small, which reduces the generalizability of these findings. Consequently, we recommend that scientists follow experimental procedures used in single studies that utilized their target population. Third, the use of VER following GXT likely elicits the highest estimate of VO_2_max on that day, yet it is possible that additional testing on subsequent days could elicit higher estimates of VO_2_max, as recently shown [38]. However, requiring multiple sessions of exercise including GXT and VER on many days may not be appropriate in unhealthy participants due to time and health related challenges.

## 5. Conclusions

In conclusion, results from this review demonstrate that verification testing typically leads to similar estimates of VO_2_max versus prior incremental exercise in unhealthy adults and children having a range of conditions that diminish health status and overall function. This result is informed from verification testing requiring submaximal, maximal, and supramaximal intensities, and it is apparent that each protocol is able to verify VO_2_max attainment in this particular sample. However, many participants reveal higher VO_2_max in response to VER compared to GXT, which substantiates its use when the most accurate estimate of VO_2_max is needed. Moreover, it is a safe and well-tolerated protocol that does not induce contraindications to exercise, and its only shortcoming is the additional time required of the participant. It is evident that some individuals do show higher VO_2_max in response to verification testing. This merits implementation of this additional test when detecting small differences in VO_2_max are paramount, for example, to identify potential health risks or describe the efficacy of exercise training in specific clients to augment health status. Failure to do so may lead to inaccurate courses of treatment which may diminish health status of patient populations.

## Figures and Tables

**Table 1 sports-09-00108-t001:** Summary of studies included in this review.

Study	Participants	Age (Years)	BMI (kg/m^2^)	Physical Activity Classification
Leicht et al. [24]	24 M wheelchair athletes	28 ± 6	NR	Active
Frederike de Groot et al. [25]	20 children with spina bifida	10 ± 5	19 ± 4	NR
Mahoney et al. [26]	9 M with obesity	24 + 6	33 + 4	Recreationally active
Arad et al. [27]	35 Sedentary M/W	29 ± 4	NR	Inactive
Causer et al. [28]	28 M/W with cystic fibrosis	31 ± 12	22 ± 3	NR
Astorino et al. [9]	24 Sedentary M/W	22 ± 4	25 ± 2	Inactive
Astorino et al. [23]	10 M/W SCI 10 M/W AB	33 ± 10 SCI 24 ± 7 AB	23 ± 3 SCI 24 ± 3 AB	Recreationally active
Astorino et al. [13]	17 W with obesity	37 ± 10	39 ± 4	Inactive
Bhammar et al. [29]	11 M/W with hypertension	22 ± 2	24 ± 3	NR
Werkman et al. [30]	16 adolescents with cystic fibrosis	14 ± 2	18 ± 1	NR
Misquita et al. [31]	108 W who are Postmenopausal	60 ± 6	33 ± 4	Inactive
Bhammar et al. [32]	9 NO children 9 OB children	11 ± 1	18 ± 1 NO 29 ± 4 OB	NR
Bowen et al. [15]	24 M with symptomatic CHF	64 ± 11	30 ± 3	NR
Dalleck et al. [10]	18 Older M/W	59 ± 6	28 ± 3	Recreationally active
Moreno-Cabañas et al. [14]	100 M/W with metabolic syndrome	57 ± 8	32 ± 5	Inactive
Sawyer et al. [12]	19 M/W with obesity	35 ± 8	36 ± 5	Inactive
Saynor et al. [33]	13 adolescents with cystic fibrosis	13 ± 3	21 ± 4	NR
Schaun et al. [22]	33 adults with hypertension	67 ± 5	32 ± 6	NR
Schneider et al. [21]	43 W with breast cancer; 32 M with prostate cancer	61 ± 12	26 ± 4	Recreationally active
Wood et al. [11]	135 M/W with Overweight or Obesity	37 ± 5	30 ± 2	Inactive

M = men; W = women; BMI = body mass index; OB = obesity; NO = normal weight; NR = not reported; AB = able-bodied; SCI = spinal cord injury.

**Table 2 sports-09-00108-t002:** Methodological traits of exercise testing of studies included in this review.

Study	Exercise Mode	Traditional VO_2max_ Criteria Adopted	VO_2max_ Protocol	Recovery Phase Protocol	VER Protocol	VER vs. GXT Criteria
Arad et al. [24]	CE	VO_2_ plateau; RER ≥ 1.10; ≥95% HRmax	RAMP 4 min unloaded cycling + 1 W/3 s for women 1 W/4 s for men	10 min active recovery at 25 W + 2–3 min passive	100% PPO	NR
Astorino et al. [9]	CE	NR	STEP 14 W/min for women 21 W/min for men and 5 W/20 s for women and 10 W/20 s for men	1–1.5 h or 24 h later	2-min WU at 28 W for women, 42 W for men followed by cycling at 105 or 115% PPO	NR
Astorino et al. [23]	ACE	VO_2_ plateau using individual ΔVO_2_ values for each participant	RAMP 5 min warm-up + 3 W/min for TETRA, 13 W/min for PARA, and 8–20 W/min for AB	10 min active recovery at 7 W	2 min at 7 W + arm cycling 105% PPO	NR
Astorino et al. [13]	CE	NR	RAMP 40 W for 2 min + 20 W/min	10 min active recovery at 20 W	2 min WU at 20 W + cycling at 105% PPO	A conservative difference in VO_2_max between protocols <0.06 L/min was used to identify ‘true’ VO_2_max
Bhammar et al. [32]	CE	RER ≥ 1.00, HR ≥ 90% of age-predicted HRmax	STEP 6 min at 40 W + initial WR of 20 W followed by 10–15 W/min	15 min of passive recovery	2 min WU at 20 W + cycling at 105% PPO	Measured VER VO_2_max was considered higher than measured GXT VO_2_max when difference between measured VER and GXT VO_2_max was greater than the difference between predicted values
Bhammar et al. [29]	CE	HR > 85% age-predicted HRmax; RER > 1.15	STEP 40 W + 20 W/min for women 50 W + 25 W/min for men	15 min passive recovery	2 min WU at 30 W for women, 40 W for men + cycling at 105% PPO	VER-derived VO_2_max was higher than incremental VO_2_max when the difference between measured VER VO_2_max and incremental VO_2_max was greater than the difference between predicted VER and incremental VO_2_max
Bowen et al. [15]	CE	BLa > 8 mM; HR within 10% of age-predicted HRmax; RPE > 18; RER > 1.00–1.15	RAMP 4 min at 10 W + 4–18 W/min	5 min active recovery at 10 W	4 min WU at 10 W + cycling at 95% PPO	NR
Causer et al. [28]	CE	VO_2_ plateau; RPE > 9; RER > 1.03–1.05; Predicted VO_2_peak, PPO, or HRpeak	RAMP 3 min at 20 W + 10–25 W/min	5 min cool-down at 20 W +10 min seated rest	3 min WU at 20 W + cycling at 110% PPO	Less than 9% difference between protocols
Dalleck et al. [10]	CE	RER > 1.0–1.15; HR within 10 b/min of age-predicted HRmax; VO_2_ plateau	STEP2 min WU at 50 W + 10–15 W/min	60 min passive recovery	2 min WU at 50 W + cycling at 105% PPO	Less than 3% difference in VO_2_max between tests
de Groot et al. [25]	TM	Heart rate = 95% (210–age); RER > 1.0; VO_2_ plateau	STEP 2% grade + 2 km/h + 0.25% change in grade/min or 3 km/h + 0.50% change in grade per min	4 min passive recovery	110% peak speed	Difference in VO_2_max between protocols >2.1 mL/kg/min
Leicht et al. [24]	TM	VO_2_ plateau; RER > 1.05 BLa > 4.0 mM; HR > 85% age-predicted HRmax	STEP Constant speed at 1% grade and grade increased by 0.1–0.3%/min	5 min active recovery at 1 m/s at 1% grade	Same peak speed as GXT but supramaximal gradient (+0.6% for PARA and NON-SCI; +0.3% for TETRA)	NR
Mahoney et al. [26]	CE	NR	RAMP 5 min WU at 20 W before power continuously increased that was individualized for each participant	At least 2 days later	2 min rest + 5 min WU at 50 W + cycling at 80–105% PPO	NR
Misquita et al. [31]	TM	HRmax > 220–age; RER > 1.1; VO_2_ plateau	STEP Bruce protocol	1–2 min of slow walking +2 min at 0% incline at a speed eliciting 70%HRmax	Balke protocol TM grade was increased to 4% for 2 min and increased 2%/min	NR
Moreno-Cabañas et al. [14]	CE	VO_2_ plateau; RER > 1.1; BLa 8 mM; HR < 5% from age-predicted HRmax	RAMP 3-min WU at 30 W for women, 50 W for men + 15–20 W/min	5 min active recovery at 30 W + 15 min seated recovery	2 min WU at 30 W for women, 50 W for men + cycling at 110% PPO	NR
Sawyer et al. [12]	CE	NR	RAMP 5 min WU 50 W + 30 W/min for men 25 W + 15 W/min for women	Active recovery for 5–10 min at 25 or 50 W	100% PPO	NR
Saynor et al. [33]	CE	NR	RAMP 3 min at 20 W + 10–25 W/min	5 min active recovery at 20 W + 10 min passive seated recovery	3 min at 20 W + cycling at 110% PPO	NR
Schaun et al. [22]	TM	ΔVO_2_ ≤ 150 mL/min; RER > 1.10; RPE ≥ 18; ± 10 b/min of 220–age	STEP 3 min at 3 km/h + 0.5 km/h and 1% increments in speed and grade	10 min of passive recovery	2 min at 50% of peak speed/grade + 1 min at 70% peak speed/grade + exercise at 1 stage higher than GXT	Difference in VO_2_max between protocols < 3%
Schneider et al. [21]	CE	RER ≥ 1.1; HRmax ≥ 200 b/min–age BLamax ≥ 8 mM; RPE ≥ 18	STEP 20 W + 10 W/min	10 min passive recovery	cycling at 110% PPO	VO_2_max in VER does not exceed GXT-derived value by >3%
Werkman et al. [30]	CE	VO_2_ plateau; HR > 95% age-predicted HRmax; RER > 1.0	RAMP Unloaded cycling + 10 W/min < 120 cm; 15 W/min 120–150 cm; 20 W/min > 150 cm	1 min passive recovery + 1 min unloaded cycling	Test started with an increase in PO every 10 s based on each participant’s height	NR
Wood et al. [11]	TM	VO_2_ plateau; HR ± 11 b/min of age-predicted HRmax RER ≥ 1.15; BLa ≥ 8 mM; RPE ≥ 18	STEP 4 min at 5.6 km/h ^−1^ and 0% grade + increased velocity to a speed consistent with face-paced walk slow jog + 2.5% change in grade/min	5–10 min passive recovery	0.5 km/h above maximum workload in GXT achieved through increases in speed and/or grade	Change in VO_2_ < 50% of that expected for the change in mechanical work

RAMP = ramp protocol; STEP = step protocol; CE = cycle ergometry; TM = treadmill; WU = warm-up; HR = heart rate; RER = respiratory exchange ratio; VER = verification test; VO_2_ = oxygen uptake; GXT = graded exercise test; BLa = blood lactate concentration; RPE = rating of perceived exertion; ACE = arm cycle ergometry.

**Table 3 sports-09-00108-t003:** Results from studies included in this review.

Study	VO_2max_ GXT (mL·kg·min^−1^)	GXT Duration (min)	VO_2max_ VER (mL·kg^−1^·min^−1^)	VER Duration (min)	HR_max_ GXT (b/min)	HR_max_ VER (b/min)	Results
Arad et al. [27]	28 ± 6	9.6 ± 1.6	30 ± 7 *	2.6 ± 0.5	170 ± 12	172 ± 9	VER elicited a higher VO_2_peak versus GXT, although there was no difference in HRpeak.
Astorino et al. [9]	32 ± 4	10.5 ± 1.6	32 ± 5	2.7 ± 0.7	191 ± 9 *	187 ± 10	There was no difference in VO_2_max between protocols, yet several participants demonstrated a higher VO_2_max in response to VER. GXT revealed a higher HRmax versus VER.
Astorino et al. [23]	17 ± 4 SCI 24 ± 4 AB	7.4 ± 1.4	17 ± 4 SCI 26 ± 4 * AB	1.7 ± 0.3	161 ± 29 176 ± 17	160 ± 26 178 ± 12	Mean VO_2_peak from VER was higher than GXT in the AB group, although VO_2_peak was similar across protocols in SCI. There was no difference in HRpeak across all groups between protocols.
Astorino et al. [13]	2.0 ± 0.4 L/min	NR	2.0 ± 0.3 L/min	1.5 ± 0.3	174 ± 13	174 ± 12	There was no difference in VO_2_max or HRmax between protocols, although 5, 9, and 7 women revealed a verification VO_2_max > 0.06 L/min higher versus GXT.
Bhammar et al. [32]	40 ± 4 NO 27 ± 4 OB	9.7 ± 2.4	43 ± 4 * NO 28 ± 3 OB	2.2 ± 0.5	189 ± 6 NO 190 ± 13 OB	184 ± 8 NO 188 ± 12 OB	All children exhibited higher mean VER VO_2_max versus GXT, although there was no difference in HRmax.
Bhammar et al. [29]	31 ± 6	NR	32 ± 6	2.1 ± 0.3	180 ± 11	180 ± 7	There was no difference in VO_2_max or HRmax between protocols, yet 3 of 11 participants exhibited a higher VO_2_max during VER compared to GXT.
Bowen et al. [15]	14 ± 3	5.8–15.1 ± 0.5–1.9	15 ± 3	2.0 ± 0.4	117 ± 20	119 ± 26	Mean VO_2_peak and HRpeak were not different between protocols and VO_2_peak was confirmed in 60% of participants.
Causer et al. [28]	35 ± 8	9.3 ± 2.3	33 ± 7	1.5 ± 0.4	168 ± 15	NR	Mean VO_2_peak did not differ between protocols, yet VER VO_2_peak was higher than GXT in 21% of participants.
Dalleck et al. [10]	28 ± 6	10.1 ± 2.1	27 ± 6	2.5 ± 0.5	165 ± 11	164 ± 10	Mean VO_2_max and HRmax were not different between protocols, although 11% of subjects exhibited higher VO_2_max and HRmax values with VER.
Frederike de Groot et al. [25]	34 ± 8	9.0 ± 4.0	35 ± 8	NR	184 ±20	NR	Mean VO_2_peak was similar between protocols, yet 25% and 42% of participants showed a higher VO_2_peak and HRpeak in VER versus GXT.
Leicht et al. [24]	23–40 ± 3–6	8.5–10.5 ± 0.5–2.5	NR	NR	125–188 ± 7–10	125–181 ± 7–15	VO_2_peak and HRpeak did not differ between VER and GXT in all subgroups. Athletes tended to exhibit a lower VO_2_peak in response to VER versus GXT.
Mahoney et al. [26]	3.4 ± 0.4 L/min	8.3 ± 0.4	3.4–3.6 ± 0.5 L·min^−1^	2.5–6.9 ± 0.4–2.5	175 ± 12	170–177 ± 13–17	VER performed at 90% PPO elicits greater VO_2_max versus GXT, yet there was no difference in HRmax.
Misquita et al. [31]	19 ± 3	8.8 ± 1.9	20 ± 3 *	8.5 ± 1.9	156 ± 15	158 ± 14	VER revealed higher VO_2_peak versus GXT, although HRpeak was similar.
Moreno-Cabañas et al. [14]	23 ± 8	7.9 ± 2.0	25 ± 8 *	2.1 ± 0.4	155 ± 15	156 ± 15	VER-derived VO_2_peak was higher than GXT, although there was no difference in HRpeak. Forty percent of participants show underestimated VO_2_peak in response to GXT that is confirmed with VER.
Sawyer et al. [12]	2 ± 1 L·min^−1^	7.1 ± 1.9	2 ± 1 L·min^−1^	1.9 ± 0.4	174 ± 16	177 ± 13 *	Mean VO_2_max was not different between protocols, yet HRmax was higher in VER. Thirteen and 8 participants achieved a VO_2_max and HRmax in response to VER that was ≥2% and 4–14 b/min higher than GXT.
Saynor et al. [33]	34 ± 3	8–12	NR	NR	187 ± 15	NR	VO_2_max values are reproducible in this sample in response to GXT and VER.
Schaun et al. [22]	22 ± 5	12 ± 2	24 ± 6 *	4.7 ± 0.4	150 ± 16	152 ± 16	VO_2_max was higher in response to VER versus GXT, although there was no difference in HRmax.
Schneider et al. [21]	21 ± 4	13.0 ± 2.9	21 ± 5 *	2.2 ± 0.3	150 ± 20	151 ± 21	VO_2_max from VER was lower than GXT, although there was no difference in HRmax. Sixty-eight percent of participants showed a ‘true’ VO_2_max with VER, although 32% elicited a 3–21% higher VO_2_max.
Werkman et al. [30]	39 ± 7	11.0 ± 3.0	39 ± 9	4.0 ± 1.0	177 ± 12	179 ± 13	There was no difference in VO_2_peak or HRpeak between protocols.
Wood et al. [11]	34 ± 7	8–12	34 ± 7	NR	180 ± 10	180 ± 10	Neither VO_2_peak nor HRpeak were different between protocols.

VO_2_max = maximal oxygen uptake; GXT = graded exercise test; VER = verification test; HR = heart rate; NO = normal weight; OB = obese; * = *p* < 0.05 between protocols.

## Data Availability

All relevant data are presented in this paper.

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
