# Peer review of "Utility of Verification Testing to Confirm Attainment of Maximal Oxygen Uptake in Unhealthy Participants: A Perspective Review"

_sports, 2021, doi:10.3390/sports9080108_

Round 1
Reviewer 1 Report
Despite the relative interest of the article, it seems to me that there is no adequate justification for the importance of this systematic review.
Why is VO2max assessment in unhealthy people more important than in healthy people?
Why should there be a single protocol for measuring a physiological variable that can be achieved in a wide variety of exercise types?
The autors mention that there is no robust way to know if vo2max is reached? so isn't the plateau occurrence, which often occurs, a robust form?
They do not clearly indicate the advantages and disadvantages of repeating a maximum effort in a subject with health problems.
They don't clearly explain why it is so important to know so rigorously the true value of vo2max in these subjects (to better prescribe the training?!!; to better discriminate them?
How do we know that the vo2max achieved in VER is really the highest value?
What are the validation criteria used in VER? Aren't they the same?
Confirming the previous value means just that. Both values ​​may not be the "true" VO2max.!!
The VER although it can be well tolerated, it can be harmful. Requesting two maximal efforts in this population doesn't seem very adequate to me, mainly because it only serves to confirm the first value of vo2max.
The authors do not clearly state why de accuracy of 1.5ml/kg/min might be clinically significant in these populations. ... and even in others!
The authors state that VER is "relatively new method identified to perform this function". Strictly speaking, it is another protocol used for many years to determine VO2max. Ractangular protocols, of very high constant intensity (above the maximal lactate state), which exhaust the subject within 3 to 10 minutes, have been used for a long time.
The authors do not consistently justify the importance of doing this systematic review in unhealthy people.
VER is nothing more than a protocol that can be used to determine VO2max. It can confirm the value obtained in the GTX test, but it has not been proven to measure the "true" VO2max. The authors themselves state that they usually confirm the vo2max obtained in the GTX test.
The advantages and disadvantages of using it to confirm the VO2max obtained before, seem the same in any type of population. The authors do not justify whi it is of critical importance in unhealthy people.
Author Response
Please see attached responses--thank you.

Reviewer 2 Report
This manuscript was designed to evaluate the efficacy of Verification Test in order to confirm that VO2 max was attained during a part incremental test. Considering that VO2 max determination may be negatively affected by certain conditions (aging, disease, etc) that could limit the VO2 to attain a maximal value, applying the Verification Test would be a way to confirm that the participants reached VO2 max.
Major comment
Although the topic seems interesting, I suggest discussing the findings found in active and inactive individuals, as well as the risk for cardiovascular disease (obesity, hypertension, etc)
Previous studies have already shown that Verification Test appears to be effective in confirming attainment of VO2 max in the clinical population (individuals with metabolic syndrome and obesity). doi: 10.1111/sms.13602. doi.org/10.1152/japplphysiol.00455.2020. I believe that divergent findings may be reported in the literature; however, this information is missing (or not clear enough) in the Introduction Section of the manuscript. Thus, I would suggest the authors focus on discussing the efficacy of Verification Test only in the clinical population, highlighting the outcomes in Discussion Section.
Tables should bring only findings related to the clinical population in order to increase the relevance of the present study.
Finally, I suggest the authors define better the main question of the present review in order to clarify for the readers.
Minor comment
Please revise line 51
Author Response
Please see attached responses and our comments to you as Reviewer 2; thank you.

Round 2
Reviewer 1 Report
The usefulness of using a protocol other than the initial test to verify VO2max is still not justified;
The V.E.R. it is a known and used protocol for a long time in the evaluation of VO2max;
We can use several protocols to check if the VO2max of the first test shows an adequate value;
The authors do not justify why this V.E.R. has special application in unhealthy people;
Overall, the article may be of interest to confirm that we can use different protocols in VO2max testing, but it doesn't seem appropriate with this title: "Utility of Verification Testing to..."
The possibility of using different tests to assess (or verify) VO2max has been known for a long time; thus, the article does not add much value to knowledge in this area.
Author Response
We have made additional changes to the paper based on comments added by academic editors. In addition, we have added a new paper to our review based on data from hypertensive adults (Schaun et al. 2021) showing that compared to GXT, VER elicits a significantly higher value of VO2max on an aggregate and individual level. We believe that the text already added to our manuscript based on your prior comments and this new set of data substantiate our claim that VER is a necessary protocol to confirm VO2max in similar populations. This result combined with data from 5 other studies supports its efficacy and we are sorry that this does not agree with your contention.
Reviewer 2 Report
The manuscript was significantly improved. The authors met all comments satisfactorily
Author Response
Thank you for your comments.